# Copper Phosphide Nanowires as High-Performance Catalysts for Urea-Assisted Hydrogen Evolution in Alkaline Medium

**DOI:** 10.3390/ma16114169

**Published:** 2023-06-03

**Authors:** Hui Shen, Tianran Wei, Junyang Ding, Xijun Liu

**Affiliations:** 1School of Bioengineering, Hefei Technology College, Hefei 230012, China; 2State Key Laboratory of Featured Metal Materials and Life-Cycle Safety for Composite Structures, School of Resources, Environment and Materials, Guangxi University, Nanning 530004, China; 3Institute for New Energy Materials & Low-Carbon Technologies, School of Materials Science and Engineering, Tianjin University of Technology, Tianjin 300384, China

**Keywords:** energy-efficient, hydrogen energy, water electrolysis, urea oxidation reaction, catalysts design

## Abstract

Water electrolysis represented a promising avenue for the large-scale production of high-purity hydrogen. However, the high overpotential and sluggish reaction rates associated with the anodic oxygen evolution reaction (OER) posed significant obstacles to efficient water splitting. To tackle these challenges, the urea oxidation reaction (UOR) emerged as a more favorable thermodynamic alternative to OER, offering both the energy-efficient hydrogen evolution reaction (HER) and the potential for the treating of urea-rich wastewater. In this work, a two-step methodology comprising nanowire growth and phosphating treatment was employed to fabricate Cu_3_P nanowires on Cu foam (Cu_3_P-NW/CF) catalysts. These novel catalytic architectures exhibited notable efficiencies in facilitating both the UOR and HER in alkaline solutions. Specifically, within urea-containing electrolytes, the UOR manifested desirable operational potentials of 1.43 V and 1.65 V versus the reversible hydrogen electrode (vs. RHE) to reach the current densities of 10 and 100 mA cm^−2^, respectively. Concurrently, the catalyst displayed a meager overpotential of 60 mV for the HER at a current density of 10 mA cm^−2^. Remarkably, the two-electrode urea electrolysis system, exploiting the designed catalyst as both the cathode and anode, demonstrated an outstanding performance, attaining a low cell voltage of 1.79 V to achieve a current density of 100 mA cm^−2^. Importantly, this voltage is preferable to the conventional water electrolysis threshold in the absence of urea molecules. Moreover, our study shed light on the potential of innovative Cu-based materials for the scalable fabrication of electrocatalysts, energy-efficient hydrogen generation, and the treatment of urea-rich wastewater.

## 1. Introduction

The rapid proliferation of the global population, and the advancement of civilization have engendered an exponential escalation in energy demand [1,2], while fossil fuels as a predominant stakeholder in the energy sector have exacerbated a myriad of predicaments, including energy insufficiencies [3], environmental calamities [4], climate change [5], and human health issues [6]. These formidable challenges have underscored the imperative to foster renewable energy alternatives, which have been exemplified by solar power [7], wind energy [8], hydrogen [9], and other viable options. Among these alternatives, hydrogen energy exhibited exceptional promise owing to its elevated calorific value and negligible ecological footprint [10]. However, the anodic OER used in existing technology consumed a great deal of energy due to its sluggish kinetics and high overpotential [11,12]. Researchers have been seeking more favorable anodic reactions, e.g., small molecules oxidation (urea, hydrazine, alcohol, aldehyde, amine, and glucose, etc.), to replace the OER and improve the efficiency of hydrogen production [13]. Among them, one promising option is the UOR, which has an ideal theoretical thermodynamic voltage of just 0.37 V, which is considerably lower than the OER’s value of 1.23 V [14,15], indicating its potential in the purification of urea-containing wastewater as well as low energy consumption for hydrogen production.

Noble metals such as Pt, Ir, and Ru have gained recognition for their exceptional performance in UOR [16,17]. Nevertheless, their elevated cost and limited accessibility have imposed restrictions on their practical implementation. To tackle this predicament, extensive endeavors have been directed towards the development of non-precious transition metal-based catalysts for the UOR, which have exhibited both cost-effectiveness and comparable activities to their costly counterparts. Recently, there has been burgeoning interest in Ni-/Co-based materials, encompassing Sn-CoS_2_ [18], Ni-Mo-S [19], Co_2_Mo_3_O_8_ [20], NiCo_2_O_4_ [21], and Fe-Ni(OH)_2_ [22], respectively, owing to their potential in bolstering the UOR. However, the exploration of Cu-based catalysts for the UOR remains in its nascent stages. For instance, Lian et al. adeptly synthesized Cu(OH)_2_ catalysts, unveiling their remarkable performance in UORs with a potential of 1.49 V vs. RHE at 10 mA cm^−2^ [23]. In a separate investigation, Yang et al. embellished CuO nanosheets with Ni(OH)_2_ nanoparticles, attaining a current density of 10 mA cm^−2^ at a potential of 1.41 V vs. RHE [24]. Despite these substantial advancements, it is worth emphasizing that UOR electrocatalysts based on Cu_3_P have not been reported yet. Moreover, prior studies have illustrated the potential of Cu_3_P-based electrodes, including crystalline Cu_3_P phosphide nanosheets [25], hierarchical Fe(PO_3_)_2_@Cu_3_P nanotube arrays [26], and Cu_3_P weaving mesh [27], as the captivating materials for electrocatalytic HER. Consequently, the pursuit of a Cu_3_P-assisted UOR system for hydrogen production holds an immense research value.

Numerous reported catalysts for the UOR and the HER have encountered challenges associated with an inadequate electroconductivity, deficient specific surface area, and restricted activity. To address these issues, a novel strategy was employed in this investigation involving the synthesis of the Cu_3_P-NW/CF catalyst. The resulting Cu_3_P-NW/CF catalyst was subsequently evaluated for its performance in an alkaline medium for both the UOR and the HER. Benefiting from the three-dimensional network structure electrode resembling a honeycomb provided by copper foam, it has since become a widely adopted substrate. The Cu_3_P-NW/CF catalyst exhibited an enhanced conductivity and a greater quantity of active sites. Consequently, it displayed working potentials of 1.43 V and −0.06 V vs. RHE to yield a current density of 10 mA cm^−2^ for the UOR and the HER, respectively. These values were found to be close to or even better than those of previously reported catalysts. Furthermore, the catalyst demonstrated low Tafel slopes for both the UOR and the HER, indicative of favorable reaction kinetics. Of utmost significance, when utilized as both the anode and cathode in a double-electrode urea electrolyzer, the Cu_3_P-NW/CF catalyst exhibited an excellent stability and necessitated a low cell voltage of 1.79 V to attain a current density of 100 mA cm^−2^. This promising performance serves to expand the potential application of Cu-based materials in the UOR.

## 2. Experimental Section

### 2.1. Materials and Synthesis

Ammonium persulfate ((NH_4_)_2_S_2_O_8_, ≥99.8%), sodium hydroxide (NaOH, ≥99.9%), and hydrochloric acid (HCl, ≥99%) were all sourced from Alfa Aesar (Haverhill, MA, USA). Copper foam and 5% Nafion solution were sourced from Sinopharm Chemical Reagent Co., Ltd. (Shanghai, China) Sodium hypophosphite (NaH_2_PO_2_, ≥99.5%) was purchased from Macklin (Shanghai, China). The absolute ethyl alcohol was sourced from the Tianjin Chemical Reagent Factory (Tianjin, China). All reagents used were analytical reagents, and were utilized directly without any further pretreatment. Distilled H_2_O (~18 Ω) was purified using the Millipore system (Burlington, MA, USA).

Synthesis of Cu(OH)_2_-NW/CF. As in the typical synthesis procedure, 4 mmol (NH_4_)_2_S_2_O_8_ and 80 mmol NaOH were dissolved in 40 mL water to prepare solution A. Then, the CF was cleaned by sonication in acetone (15 min), 3 M HCl (15 min), H_2_O (15 min), and ethanol (15 min) to remove organic impurities as well as residual oxides from its surface. Next, a pre-cleaned CF piece (0.5 cm × 2.5 cm) was placed into the above solution A and was left to age undisturbedly at room temperature for 20 min. Last, the resulting Cu(OH)_2_-NA/CF electrode was washed thoroughly with water and ethanol before being vacuum dried at 60 °C for 12 h.

Synthesis of Cu_3_P-NW/CF. To synthesize the Cu_3_P-NW/CF catalyst, the NaH_2_PO_2_ (0.1 g), and Cu(OH)_2_-NW/CF were put in two separate porcelain boats. The phosphorus source was positioned on the upstream side of the tube furnace. Then, the two precursors were heated at 350 °C for 2 h in an N_2_ atmosphere (10 sccm, standard cubic centimeters per minute) at a heat-up rate of 1 °C min^−1^ to obtain the Cu_3_P-NW/CF catalyst.

Synthesis of Cu_3_P-P/CF. To construct Cu_3_P powder, a similar phosphatization procedure was applied, except that commercial Cu(OH)_2_ powder was used as a substitute. After that, 5 mg Cu_3_P powder was dissolved in 500 μL solution (50 μL 5% Nafion, 450 μL EtOH) and treated with ultrasound for 30 min. A 100 μL ink was then dropped onto the CF (loading amount: 1.0 mg cm^−2^) and dried at room temperature to obtain Cu_3_P-P/CF control materials.

### 2.2. Characterizations

X-ray diffraction (XRD) was conducted by applying a Rigaku D/max-2200 instrument (Rigaku, Tokyo, Japan) with operating conditions of 40 kV as well as 40 mA. It was equipped with ceramic monochromatized Cu Kα radiation (λ = 0.15406 nm). Scanning electron microscopy (SEM) images were obtained using a ZEISS VLTRA-55 instrument (ZEISS, Oberkochen, Germany), and relevant energy dispersive X-ray (EDX) data were collected using a Horiba EDX system integrated into the device. Meanwhile, both transmission electron microscopy (TEM) and high-resolution TEM (HRTEM) figures were acquired using a JEM-2010 HR instrument (JEOL Ltd., Tokyo, Japan) for analyzing the morphological features. Moreover, X-ray photoelectron spectroscopy (XPS) data were obtained using a Kratos Axis Ultra DLAD X-ray photoelectron spectrometer (Kratos, Manchester, UK) with Al Kα radiation (λ = 0.834 nm) to analyze the surface elemental composition.

### 2.3. Electrochemical Tests

For all electrochemical experiments, a CHI 760 E electrochemical workstation was operated in a three-electrode cell in 1.0 M KOH and 1.0 M KOH + 0.33 M urea aqueous solution. In this system, a Hg/HgO electrode (1.0 M KOH filling solution), a graphite rod (6 mm diameter), and a modified copper foam (1 cm^2^ working area) acted as the reference electrodes, the counter electrode, and working electrodes, respectively. Before collecting the electrochemical data, several cyclic voltammetry (CV) curves were scanned until the signal was deemed to be stable. Linear sweep voltammetry (LSV) curves were obtained with the potential between 0.0~−0.6 or 1.0~2.0 V vs. RHE with a scan rate of 5 mV s^−1^. Electrochemical impedance spectra (EIS) data were acquired with an applied amplitude of 5 mV and a frequency range from 100 kHz to 0.1 Hz. All reported potentials from the electrochemical tests were then converted to the RHE based on the reported equation: E_vs. RHE_ = E_vs. Hg/HgO_ + 0.059pH + 0.098 in 1.0 M KOH, or 1.0 M KOH + 0.33 M urea electrolytes.

## 3. Results and Discussion

The synthesis process of the Cu_3_P-NW/CF electrode was depicted in Figure 1a and Appendix A, while detailed information was available in the Experimental Section. Initially, Cu(OH)_2_ nanowires were created using a chemical bath growth method and were immediately immobilized onto the copper foam. Subsequently, these nanowires underwent a phosphating treatment, resulting in the conversion into the novel Cu_3_P composition, thereby achieving the fabrication of the Cu_3_P-NW/CF electrode. SEM images of the Cu(OH)_2_-NW/CF and Cu_3_P-NW/CF electrodes were depicted in Figure 1b,c, respectively. Strikingly, both electrodes manifested uniformly dispersed nanowire arrays on their surfaces, presenting a sharp contrast to the pristine CF electrode distinguished by its sleek surface. Remarkably, no fractures or detachment of the nanowire arrays were discerned, thus accentuating the heightened rigidity of adhesion between the nanowires and CF interfaces.

The formation of the Cu_3_P-NW/CF nanowires was further verified by the TEM image in Figure 1d. The HRTEM in Figure 1e revealed clear lattice fringes with a d-spacing of 0.20 nm, which can be indexed to the crystal planes of Cu_3_P (300). The selected area electron diffraction (SAED) patterns in Appendix A also confirmed the high crystallinity of the Cu_3_P phase in the Cu_3_P-NW/CF electrode. Meantime, the elemental mapping images in Figure 1f manifested a uniform distribution of the Cu and P elements across the nanowires.

In the XRD patterns presented in Figure 2a and Appendix A, the diffraction peaks observed in the Cu(OH)_2_-NW/CF and Cu_3_P-NW/CF electrodes could be accurately matched with the simulated patterns of Cu(OH)_2_ (JCPDS No. 13–0420) and Cu_3_P (JCPDS No. 71–2261), respectively. Moreover, XPS analysis was employed to further characterize the surface elemental compositions and electronic properties of the as-prepared Cu-based electrodes. The XPS survey spectrum for the Cu_3_P-NW/CF electrode was shown in Figure 2b, providing evidence for the existence of both the Cu and P elements. The high-resolution XPS spectrum of Cu 2*p* and P 2*p* in the Cu_3_P-NW/CF electrode were shown in Figure 2c,d, respectively. In the high-resolution Cu 2*p* spectrum, two distinct peaks at 933.9 eV and 953.5 eV were observed, ascribing to its 2*p*_3/2_ and 2*p*_1/2_ region, thereby revealing the presence of the Cu–P bond (Figure 2c) [28,29]. Furthermore, a broad peak centered around 129.5 eV in the P 2*p*_3/2_ region, providing compelling evidence for the formation of the Cu_3_P phase (Figure 2d) [30]. However, besides the Cu-P bonds, a new discovery was made regarding the presence of the Cu-O bond in the high-binding-energy region of the Cu 2*p* spectrum, with two peaks at 934.9 eV (2*p*_1/2_) and 955.5 eV (2*p*_3/2_), respectively. This suggested that the partial oxidation of the Cu_3_P nanowire surface formed as a result of the exposure to air, which was further supported by the noticeable P-O peak [31,32]. Additionally, the Cu 2*p* spectra exhibited three peaks at binding energies of 940.7, 943.4, and 961.8 eV, corresponding to the satellite peaks of Cu 2*p*_3/2_ and Cu 2*p*_1/2_, respectively [33,34]. Collectively, these XPS findings offer valuable insights into the elemental compositions and electronic states of the Cu_3_P-NW/CF electrode.

The electrochemical performance of the Cu_3_P-NW/CF electrode was subjected to a comprehensive investigation in two distinct electrolytes for both the UOR and the OER: 1.0 M KOH + 0.33 M urea and 1.0 M KOH (Appendix A). The purpose of this extensive analysis was to unravel the catalytic behavior of the Cu_3_P nanowire arrays during the UOR. Remarkably, the incorporation of the Cu_3_P nanowires into the CF framework led to a significant augmentation in catalytic activity, as evidenced by the striking disparity between Cu_3_P-NW/CF and CF in terms of UOR performance (Appendix A). This notable improvement underscored the pivotal function played by the Cu_3_P nanowires in facilitating the UOR process. Meantime, the Cu_3_P-P/CF electrode was also subjected to scrutiny as a control specimen (Appendix A), imparting valuable insights into the UOR behavior. The LSV curves presented in Figure 3a were also analyzed, notably revealing that the Cu_3_P-NW/CF electrode required higher potentials for the OER (1.50 V vs. RHE) compared to that of the UOR (1.43 V vs. RHE) to achieve a current density of 10 mA cm^−2^. This observation suggested a more favorable electrochemical landscape for the UOR process, indicating a promising pathway for efficient hydrogen production. Furthermore, the Cu_3_P-NW/CF electrode demonstrated a far superior UOR performance compared to the Cu_3_P-P/CF electrode, exemplifying the beneficial effects of the ordered in situ nanowire arrays generated on the conductive CF substrate. This enhanced UOR performance further underscores the role of the unique architecture in significantly boosting the electrocatalytic activity [35]. To delve deeper into the electrochemical kinetics, the Tafel slopes for the OER and the UOR of the catalysts were examined (Figure 3b and Appendix A). The Tafel slopes for Cu_3_P-NW/CF were found to be 81.3 mV dec^−1^ for the OER and 33.1 mV dec^−1^ for the UOR, respectively. These values indicated accelerated UOR kinetics compared to OER kinetics. In contrast, the Cu_3_P-P/CF electrode exhibited similar Tafel slope trends for both the OER and the UOR, suggesting a distinct advantage of the self-supported well-ordered architecture over conventional powder materials [36]. The unique architecture of the Cu_3_P-NW/CF electrode not only provided an abundance of active sites [37], but also facilitated the efficient transport of electrons and ions at the electrode–electrolyte interface [38].

Based on the widely used double layer capacitance (*C*_dl_) test, these samples’ electrochemical active surface areas (ECSA) were also assessed since a positive correlation was found to exist between them [39,40]. As depicted in Figure 3c, the Cu_3_P-NW/CF electrode had a *C*_dl_ value of 7.5 mF cm^−2^, which was approximately 1.4 times higher than the value of the Cu_3_P-P/CF electrode (5.3 mF cm^−2^). These results suggest that the construction of the self-supported nanowire arrays through an in situ method can effectively improve the catalyst’s active area [41], accelerate the diffusion of electrolytes [42], and promote the release of gaseous product bubbles from the active sites [43]. Moreover, the EIS was tested to further investigate the charge-transfer kinetic features of the Cu_3_P-NW/CF and Cu_3_P-P/CF electrodes. Figure 3d revealed that the Cu_3_P-NW/CF electrode had a smaller solution resistance and ion diffusion resistance compared to the Cu_3_P-P/CF electrode, demonstrating its fast reaction kinetics and contributing to its excellent UOR activity. The chronoamperometry method provided compelling evidence of the excellent stability and durability of Cu_3_P-NW/CF in alkaline electrolytes comprising 1.0 M KOH + 0.33 M urea. As shown in Figure 3e, there is an almost imperceptible attenuation for the current density of ~77 mA cm^−2^ after 30 h continuous operation at 1.6 V vs. RHE. The SEM images captured before and after the stability tests indicated no discernible alterations in the structure and morphology of the nanowire arrays (Appendix A), further substantiating their remarkable robustness.

The HER capability of the Cu_3_P-NW/CF electrode was further evaluated in a 1.0 M KOH aqueous solution and compared with the Cu_3_P-P/CF electrode and Pt/C under identical conditions (Figure 4a). Undoubtedly, within the examined voltage window, the cathode current of individual CF was found to be virtually nonexistent in contrast to the benchmark Pt/C, which demonstrated the highest activity. At universal current densities of −10 and −100 mA cm^−2^, the Cu_3_P-NW/CF exhibited exceptional catalytic activity, demanding low overpotentials of 60 and 389 mV, respectively. Conversely, at the equivalent current densities, the Cu_3_P-P/CF electrode was capable of acquiring high overpotentials of 170 and 583 mV, respectively (Figure 4b).

In Figure 4c, the computed Tafel slope for the Cu_3_P-NW/CF hybrid was 139.5 mV dec^−1^, which was lower than that of the Cu_3_P-P/CF catalyst (189.1 mV dec^−1^), but higher than that of Pt/C (39.7 mV dec^−1^), demonstrating a higher HER catalytic reaction kinetic rate for Cu_3_P-NW/CF. In comparison, the Cu_3_P-P/CF powder-based electrode exhibited comparatively sluggish kinetics due to its detrimental structural characteristics that alleviated the HER on the catalyst surface [44]. The HER process at Cu_3_P-NW/CF followed a Volmer–Heyrovsky mechanism [45,46], with the electrochemical desorption of adsorbed hydrogen being considered as the rate-determining step [47,48]. To assess durability, a chronoamperometric test was conducted at overpotentials of −0.7 V vs. RHE for 30 h. Figure 4d depicts the continuous functioning of the Cu_3_P-NW/CF at the current density of ~102 mA cm^−2^ in 1.0 M KOH aqueous electrolyte for 30 h.

The exceptional UOR and HER achievement of the Cu_3_P-NW/CF catalyst prompted an investigation into its potential application in energy-efficient hydrogen generation systems through urea electrochemical oxidation. A two-electrode urea-assisted electrolyzer was assembled, using both the working electrode and the counter electrode consisting of Cu_3_P-NW/CF. Figure 5a showed that, in contrast to the Cu_3_P-P/CF electrode (1.56 V and 2.01 V), the Cu_3_P-NW/CF electrode required unusually lower operating potentials of 1.50 and 1.79 V to easily attain current densities of 10 and 100 mA cm^−2^, respectively. Furthermore, as presented in Appendix A, in the same electrolytic water device, the electrolyte containing 1.0 M KOH + 0.33 M urea was replaced with 1.0 M KOH alone. The Cu_3_P-NW/CF electrocatalyst subsequently required 1.81 V to meet 10 mA cm^−2^, which was 0.31 V higher than that under the urea-assisted system, implying a 20.7% reduction in the driving voltage for hydrogen generation. This demonstrated the cost-effectiveness of the urea-assisted hydrogen production strategy [49,50,51,52]. Additionally, the urea-assisted electrolytic device equipped with the two electrodes of Cu_3_P-NW/CF manifested a remarkable stability with a consistent cell current of approximately 100 mA cm^−2^ over a 30 h period (Figure 5b). Furthermore, its activity and stability were either close to or better than other reported metal-based catalysts in the aspect of energy-saving hydrogen production by urea electrocatalysis (Appendix A). Moreover, the SEM images of the post-stability test revealed that the nanowire arrays remained largely intact without any signs of corrosion or detachment (Appendix A). This observation underscored its excellent structural stability even after prolonged operation. Meanwhile, a comparison of the XPS spectra before and after stability testing demonstrated that the chemical composition and electronic states of the materials remained consistent with the original Cu_3_P-NW/CF electrode (Appendix A). These findings provided further evidence of the catalysts’ robust stability, and thereby affirmed their suitability for long-term applications.

The outstanding durability and activity of the Cu_3_P-NW/CF catalyst can be attributed to several factors elucidated by the aforementioned findings. First, the method utilized for growing ordered nanowire arrays in situ on CF does not require the addition of binders, thereby enhancing its stability as well as the electron transfer ability across the catalyst and the substrate in an electrochemical process. This binder-free configuration ensures robustness and mitigates the possibility of performance degradation over time. Furthermore, the unique structure of the Cu_3_P nanowire arrays provides a higher density of active sites, thereby enabling enhanced catalytic activity. At the same time, the nanowire architecture facilitates the efficient removal of generated gases through its outermost layer, preventing gas bubble accumulation and reducing mass transport limitations. This advantageous feature distinguishes the Cu_3_P-NW/CF catalyst from the conventional powder catalysts with flat surfaces, leading to improved catalytic performances and increased reaction efficiency. These combined attributes of the Cu_3_P-NW/CF catalyst, including its binder-free construction, high surface area, and efficient gas removal mechanism, contribute to its exceptional durability and activity in the urea electrochemical oxidation and hydrogen generation processes.

## 4. Conclusions

In summary, a Cu_3_P nanowire array was synthesized using a simple two-step methodology. The resulting catalyst demonstrated exceptional catalytic prowess for the UOR and the HER owing to the heightened conductivity and augmented active sites. Of particular significance, a urea-assisted electrolysis system employing dual Cu_3_P-NW/CF electrodes achieved a current density of 100 mA cm^−2^ with an impressively low applied voltage of 1.79 V, surpassing both the comparative sample and the conventional alkaline water electrolyzer. This configuration holds considerable potential for energy-efficient hydrogen production. This investigation unveils an innovative approach for the intelligent fabrication and modification of Cu-based electrocatalysts, highlighting a specific focus on hydrogen generation and other energy-related applications. Moreover, the cost-effective and highly catalytic Cu_3_P-NW/CF bifunctional electrocatalyst not only facilitates the ecologically sound treatment of urea-containing wastewater, but also enables the development of low-pressure, high-efficiency hybrid water electrolysis systems. Furthermore, it engenders numerous prospects for advancing Cu-based catalysts in the realm of electrochemistry, encompassing oxygen reduction, CO_2_ reduction, and oxidation reactions involving other diminutive molecules.

## Figures and Tables

**Figure 1 materials-16-04169-f001:**
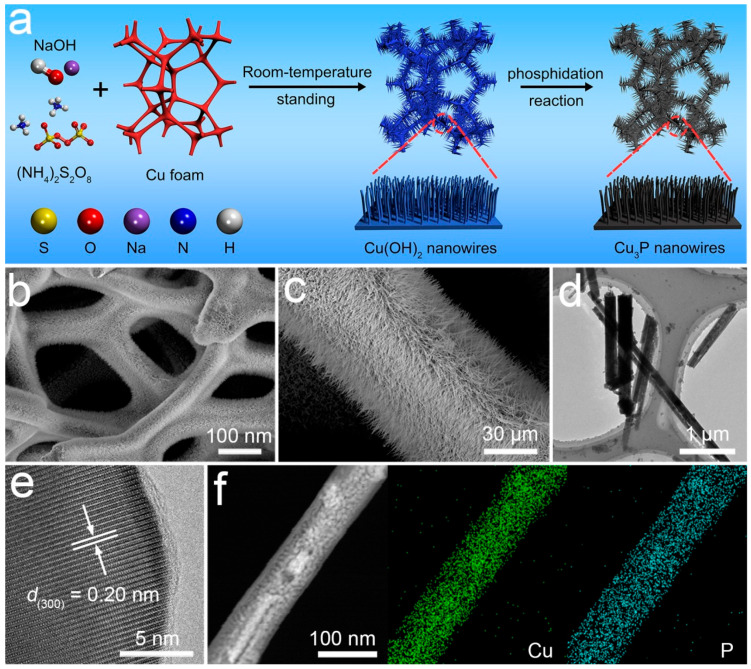
(**a**) Schematic illustration of the synthesis procedure of the Cu_3_P-NW/CF electrode. (**b**) SEM image of Cu(OH)_2_-NW/CF. (**c**–**f**) SEM, TEM, HRTEM and EDX elemental mapping images of the Cu_3_P-NW/CF electrode.

**Figure 2 materials-16-04169-f002:**
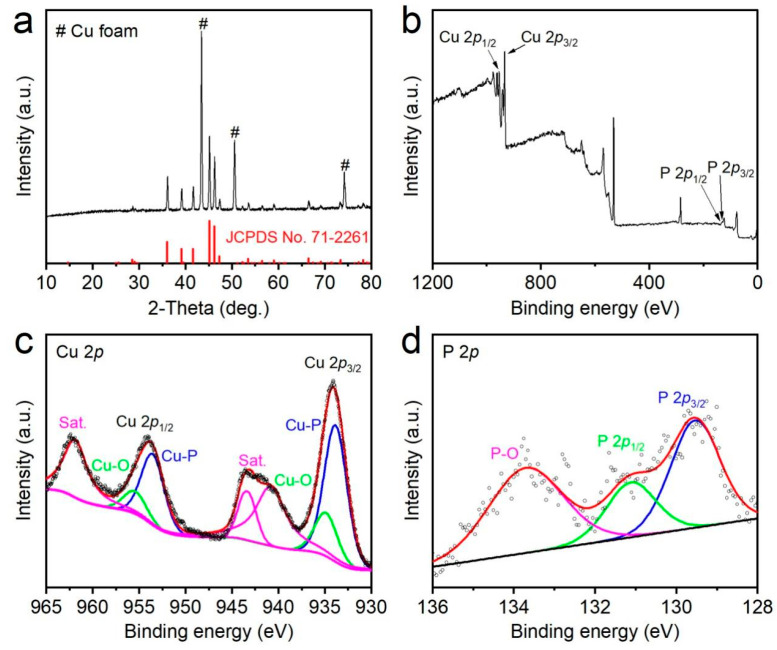
(**a**) XRD pattern of the Cu_3_P-NW/CF electrode. (**b**) XPS survey spectrum of the Cu_3_P-NW/CF electrode. (**c**,**d**) High resolution (**c**) Cu 2*p*, and (**d**) P 2*p* spectra in the Cu_3_P-NW/CF electrode.

**Figure 3 materials-16-04169-f003:**
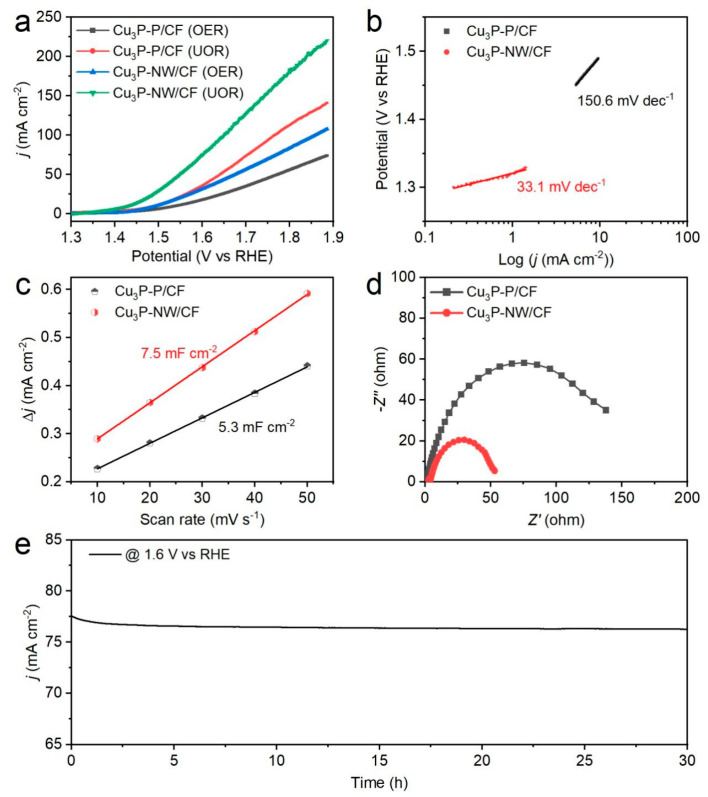
(**a**) OER/UOR polarization curves in different electrolytes composed of 1.0 M KOH and 1.0 M KOH + 0.33 M urea. (**b**) The corresponding UOR Tafel slopes. (**c**) Calculated *C*_dl_ for the two samples. (**d**) EIS spectra. (**e**) UOR stability test of the Cu_3_P-NW/CF electrode.

**Figure 4 materials-16-04169-f004:**
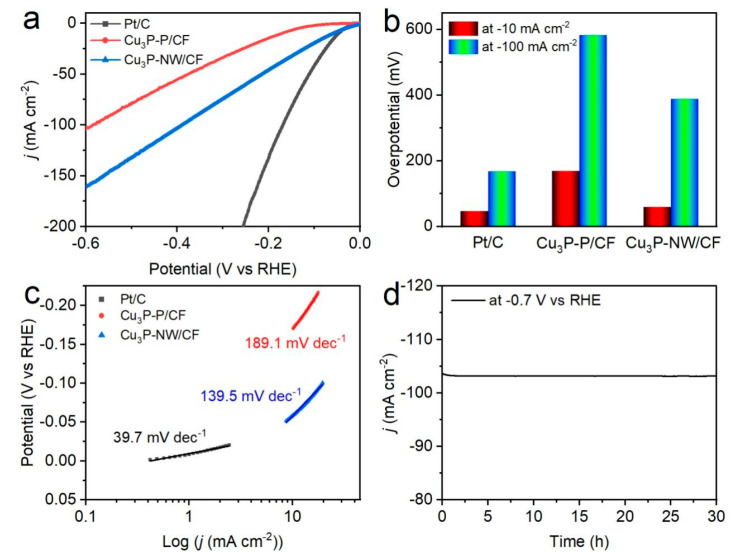
(**a**) HER polarization curves recorded in a 1.0 M KOH solution. (**b**) The performance comparison of the overpotentials required to obtain current densities of −10 and −100 mA cm^−2^. (**c**) The corresponding HER Tafel slopes. (**d**) HER stability test of the Cu_3_P-NW/CF electrode.

**Figure 5 materials-16-04169-f005:**
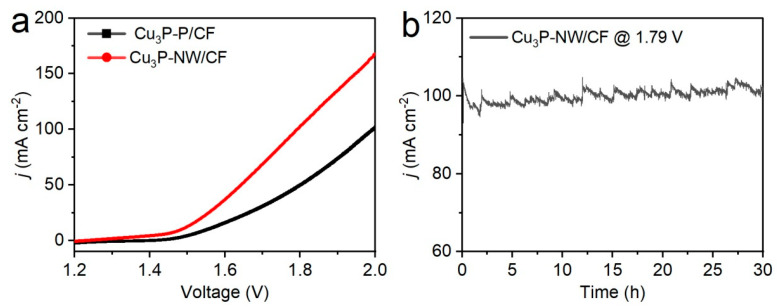
(**a**) Polarization curves of the Cu_3_P-P/CF and Cu_3_P-NW/CF electrodes for two-electrode urea electrolysis in 1.0 M KOH + 0.33 M urea (without iR correction). (**b**) Stability test of the Cu_3_P-NW/CF catalyst for urea electrolysis.

## Data Availability

Data will be made available on request.

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
