# Peer review of "Copper Phosphide Nanowires as High-Performance Catalysts for Urea-Assisted Hydrogen Evolution in Alkaline Medium"

_materials, 2023, doi:10.3390/ma16114169_

Round 1

Reviewer 1 Report

The English language should be thoroughly inspected, as should the document for grammatical errors.

The Abstract should contain answers to the following questions: What problem was studied and why is it important? What methods were used? What are the important results? What conclusions can be drawn from the results? What is the novelty of the work and where does it go beyond previous efforts in the literature? Please include specific and quantitative results in your Abstract, while ensuring that it is suitable for a broad audience. References, figures, tables, equations and abbreviations should be avoided.

The position of your citations is completely wrong. Go through the manuscript carefully from the beginning.

Avoid using personal pronouns. Personal pronouns have no place in article writing.

The problems identified in the literature and the core academic problems addressed in this manuscript should be more clearly explained in the last paragraph of the "Introduction". The motivation and novelty of the manuscript should be further addressed. The core academic problems addressed in this manuscript should be more clearly explained.

Dont use lump reference [5-9] and so on.

The writing is strongly recommended to be improved. (There is no uniformity in the manuscript). You must have to follow some recent articles to improve the structure. The abstract and Introduction section has poorly written. The introduction section should be updated with the motivation behind this study, research gap, novelty, and contributions of the work. The motivation towards the proposed work is not clear.

Some abbreviations are missing and not mentioned in the nomenclature table. Please list all abbreviations in the respective table (Nomenclature table) to reduce the length of the article, which will be at the beginning of the article after the abstract and keywords. The description of each symbol (e.g., after each equation) may be avoided if a Nomenclature is provided, otherwise, all symbols should be clearly defined at the first instance of appearance in the manuscript. The nomenclature has to include the proper sections: Latin Symbols, Greek Symbols, Subscripts and Superscripts, and Abbreviations. Put the symbols in the proper alphabetical order in every section. Also, provide the proper units for every parameter.

Regarding the discussion of the results, authors should not only state what is shown but it should be made clear to readers why the figure has been included and what is of interest. Please provide physical interpretations and insights to justify and explain the observed trends. Please expand the obtained results by adding more results with the need to confirm them by comparing them with what is currently in the literature, with the emphasis on providing sufficient scientific explanations for them.

Conclusion section is missing some perspective related to the future research work.

The English language should be thoroughly inspected, as should the document for grammatical errors.

Tenses

Personel pronouns

Author Response

Please see the attach file.

Reviewer 2 Report

Title: Copper phosphide nanowires as high-performance catalysts for 
urea-assisted hydrogen evolution in alkaline medium

The research work is interesting and useful to the production of hydrogen from 

 water electrolysis process. It may help to produce hydrogen efficiently. 

After gone through the manuscript, the following observations are made for the betterment of the research paper:

1.         It is advised to provide the schematic diagram and original photo of synthesis process.

2.         It is also advised to provide schematic diagram and original photo of electrochemical tests.

3.         The heading of Conclusion section should be “Conclusions” since there are more than one conclusion.

4.         Similarly, the heading of Reference section should be “References”.

Good

Author Response

Please see the attach file.

Reviewer 3 Report

This paper is about synthesizing Cu3P on a nickel foam and using it as an electrode for OER and UOR reactions. The synthesis method is well known and lacks originality compared to other papers. The only distinguishing point is that the developed electrode was evaluated for UOR. Therefore, I think the originality of this paper is low, but I suggest a major review based on its distinctiveness in application.

1) The author must present SEM and XPS analysis after conducting a 24-hour durability evaluation of the UOR and OER reactions at 100 mA/cm2

2) please add the SAED patterns of Cu3P.

3) The author shoud present the iR-corrected and no-iR-corrected polarization curves of two electrolyzer test. I think Figure 5 is ir-corrected. iR correction should not be applied to the two electrolyzer.

good

Author Response

Please see the attach file.

Round 2

Reviewer 1 Report

N/A